# Low-Cost Automated Vectors and Modular Environmental Sensors for Plant Phenotyping

**DOI:** 10.3390/s20113319

**Published:** 2020-06-11

**Authors:** Stuart A. Bagley, Jonathan A. Atkinson, Henry Hunt, Michael H. Wilson, Tony P. Pridmore, Darren M. Wells

**Affiliations:** 1Integrated Phenomics Group, School of Biosciences, University of Nottingham, Sutton Bonington Campus, Sutton Bonington LE12 5RD, UK; stuart.bagley1@nottingham.ac.uk; 2Future Food Beacon, School of Biosciences, University of Nottingham, Sutton Bonington Campus, Sutton Bonington LE12 5RD, UK; jonathan.atkinson@nottingham.ac.uk (J.A.A.); michael.wilson@nottingham.ac.uk (M.H.W.); 3School of Computer Science, University of Nottingham, Nottingham NG8 1BB, UK; psyhh2@nottingham.ac.uk (H.H.); psztpp@exmail.nottingham.ac.uk (T.P.P.)

**Keywords:** phenotyping robots, 3D printing, phenomics vectors, IoT sensors

## Abstract

High-throughput plant phenotyping in controlled environments (growth chambers and glasshouses) is often delivered via large, expensive installations, leading to limited access and the increased relevance of “affordable phenotyping” solutions. We present two robot vectors for automated plant phenotyping under controlled conditions. Using 3D-printed components and readily-available hardware and electronic components, these designs are inexpensive, flexible and easily modified to multiple tasks. We present a design for a thermal imaging robot for high-precision time-lapse imaging of canopies and a Plate Imager for high-throughput phenotyping of roots and shoots of plants grown on media plates. Phenotyping in controlled conditions requires multi-position spatial and temporal monitoring of environmental conditions. We also present a low-cost sensor platform for environmental monitoring based on inexpensive sensors, microcontrollers and internet-of-things (IoT) protocols.

## 1. Introduction

Plant phenotyping—the assessment of complex plant traits (architecture, growth, development, physiology, yield, etc.) and quantification of parameters underlying those traits [1,2,3]—is a rapidly developing transdiscipline of vital importance when addressing issues of global food security [4,5]. High-throughput phenotyping in controlled environments (growth chambers and glasshouses) is often delivered via large, expensive installations, leading to limited access and an increased relevance of “affordable phenotyping” solutions [6,7]. The availability of low-cost microcontrollers and automation components developed for the Maker community, combined with the ease of fabrication of 3D-printed parts allows low-cost, flexible phenotyping vector platforms to be designed for more widespread adoption [8]. We present two robotic vectors that carry sensors for plant phenotyping under controlled conditions—a linear actuator to position a thermal camera and a plate imaging robot designed to carry an RGB camera to image plate-grown plants such as the model species *Arabidopsis thaliana*. Each vector is designed for a specific task and is of inexpensive, modular construction, allowing re-design and re-purposing for other phenotyping activities as necessary. 

When phenotyping in controlled conditions, spatial and temporal environmental sensor data are essential for correct interpretation of results [9]. We also present a low-cost sensor platform for monitoring environmental conditions over a range of phenotyping setups based on inexpensive sensors, microcontrollers and internet-of-things (IoT) protocols. 

## 2. Automated Vectors

Both vectors are based on the “belt-and-pinion” linear drive principle, whereby a motor mounted on a wheeled carriage drives a timing belt that passes over the timing pulley and under the carriage wheels. The wheels then act as idler pulleys to prevent the belt losing tension (Figure 1). For increased torque and positional accuracy, a stepper motor is employed to propel the carriage and payload along a rigid drive rail. This simple configuration allows longer travel lengths and rapid carriage movement compared to leadscrew designs.

This design has been developed and adopted by the Maker community for home-built computer numerical control (CNC) machines and plotters [10] and compatible parts are readily available. A list of components used in each design is given in Table 1. Custom parts are 3D printed to reduce cost and allow flexibility and re-configuration for alternative sensor payloads or additional deployment modes. Files for all 3D-printed components are available at https://github.com/UoNMakerSpace/. All parts were printed using a fused filament fabrication 3D printer (Model S5, Ultimaker) using tough polylactic acid (PLA) filament. 

Both designs utilize microcontrollers to generate the signals to the drivers that control the stepper motor—these controllers also provide input/output signals for limit switches used as both emergency stops and home sensors. The microcontrollers themselves also trigger, configure and collect data from the sensor and provide a user-friendly interface to set experimental acquisition parameters. Microcontroller sketches and control software examples are available at https://github.com/UoNMakerSpace/.

### 2.1. Thermal Imager

The Thermal Imager is a simple linear robot designed to position a thermal camera (FLIR A35 (60 Hz)) over the canopies of plants grown in trays or pots on a standard controlled environment room shelf (Figure 2). High-throughput top-view imaging of plants can be used to measure morphological properties, such as shape and size and how these parameters develop over time [11]. The use of thermal sensors enables the measurement of physiological processes such as stomatal function [12] and responses to disease [13]. With a 19 mm lens, the field of view of the sensor is approximately 220 × 300 mm when mounted 80 cm above the canopy to be imaged. 

#### 2.1.1. Mechanical Components

The Thermal Imager comprises a horizontally-arranged aluminium carriage rail (V-slot profile, OpenBuilds) onto which is located a wheeled carriage assembled from two carriage plates. The carriage plates are 3D-printed parts with mounting holes for a NEMA17 bipolar stepper motor on one plate and a sensor attachment fitting on the other. Guide wheels are mounted between the plates and locate in the slot of the carriage rail (see Figure 1). The carriage rail is mounted on two supports fabricated from the same aluminium profile but any sturdy support will suffice. The use of aluminium profile allows easy adjustment of both carriage rail height (to adjust the sensor field-of-view) and orientation of the carriage rail (for example to a side-imaging mode to allow use with non-rosette species such as wheat, rice and barley). Files for the 3D-printed components are available at: https://github.com/UoNMakerSpace/thermal-imager-hardware. 

#### 2.1.2. Electrical/Control Components

The motor control system is based on a microcontroller development board (Arduino Uno R3)—this incorporates a 16 MHz ATmega328P controller on an inexpensive breakout board with multiple input/output connections including a USB serial connection to a host PC or laptop [14]. An expansion shield (CNC Shield V3) is connected to the board to allow deployment of up to three stepper motor drivers in the widely-used “StepStick” format [15] and multiple limit switches. The motor driver selected for this system (DRV8825, TI) can be configured to single stepping, 1/2, 1/4, 1/8, 1/16 or 1/32 microsteps and operates at a maximum drive current of 2.5 A at 24 V. Two unipolar Hall-effect sensors are wired to the shield and fixed at either end of the carriage plate. The sensors are triggered by magnets fixed to the carriage rail to act as home and limit switches. All electronic components are housed in a 3D-printed case with connectors for the stepper motor, Hall-effect sensors and motor power. The motor is powered by a 24 V, 2.71 A power adaptor. A full wiring schematic is given as Appendix A.

The microcontroller board is powered by a USB connection to the host computer, which also provides serial communication. 

#### 2.1.3. Software

The microcontroller runs a sketch written in the Arduino Integrated Development Environment [16] that uses the AccelStepper library [17] to control the stepper motor. This sketch allows setting of acceleration parameters for the motor, reads the state of the two limit switches and monitors the serial connection. The limit switch at the furthest extent of travel is an emergency stop, with the other sensor acting as a home switch—on triggering, it moves the carriage until the sensor is no longer active and sets the final position as zero (“home”). On receiving a serial string with positional information via the USB port, the carriage is moved to that position using the pre-defined acceleration parameters to ensure a smooth acceleration and deceleration before stopping and acquiring an image. Experimental parameters are set and the imaging sensor controlled by a program written in the LabVIEW development environment [18] running on the host computer. This provides a user-friendly graphical interface for control of the vector (distances moved, time-lapse parameters, etc.) and imaging sensor (Figure 3). The microcontroller sketch and LabVIEW software are available at https://github.com/UoNMakerSpace/themal-imager-software. Once acquired, image sets are processed for leaf temperature values at multiple points on each rosette using macros written for the ImageJ/FIJI image analysis platforms [19,20].

#### 2.1.4. Performance and Results

Operating characteristics of the Thermal Imager are given in Table 2. For comparison, characteristics of a previously published research system [21] and a commercially available actuator are also given. A standard experimental run with five imaging positions along the travel distance and a microstepping size of 4 is completed in 38 s including the homing sequence (which runs at each timepoint to improve repeatability). These settings give a positional accuracy of ~500 µm during extended running, no measurable discrepancy in positioning was found. We estimate the repeatability of positioning at ~5 µm. Compared to the CPIB Imaging Robot [21], the Thermal Imager has improved repeatability, speed, and temporal resolution, completing each imaging run in less than half the time. This can be attributed to the use of microstepping by the Thermal Imager driver board—the Imaging Robot does not use microstepping (cost-effective drivers were not available at the time of design), which impacts resolution and repeatability. Despite the components costing only 20% of those used in the Imaging Robot, the Thermal Imager design is thus an improvement in all operating characteristics. A typical commercial actuator (Table 2) operating at the highest level of microstepping outperforms the Thermal Imager in terms of speed and temporal resolution but at the cost of positional accuracy and hence repeatability. Importantly, current commercial systems of this specification are relatively expensive, in this case nearly 15 times more expensive than the system presented here.

An example output from the Thermal Imager is shown in Figure 4.

### 2.2. Plate Imager

The Plate Imager is designed for the automated high-throughput imaging of plate-grown plants in a standard growth room (Figure 5). Rather than continuous operation, it was designed for users to bring multiple plates for imaging at discrete time points. This approach allows different users to image many hundreds of plants using a single shared machine. With this in mind, the design focuses on throughput rather than absolute positional accuracy. Once acquired, images are processed for root system architectural traits using the RootTrace and RootNav analysis software suites [22,23,24].

#### 2.2.1. Mechanical Components

Using a similar drive system to the Thermal Imager, the Plate Imager is composed of a carriage rail on which a belt and pinion-driven carriage translocates a machine vision RGB camera (Stingray, AVT). The rail is 2 m in length, giving a working travel of 1.8 m, and allowing 14 standard 125 mm square plates to be imaged in a run. The carriage plate assembly (Figure 5b) is made from 3D-printed components and consists of a carriage plate to which is connected a sensor holder. When imaging plates, reflections from the plate lid often obsure details—to lessen this effect, a baffle plate is fitted over the front of the carriage with a cut-out for the imaging lens. This is covered in blackout material to remove reflections from the lid of the plate. Plates are mounted using 3D-printed clips against a aluminium profile bracket (covered in blackout material to provide contrast to plant roots). The carriage rail is mounted to a free-standing frame constructed from aluminium profile with an LED lighting array mounted above the drive rail to provide imaging illumination. Files for the 3D-printed components are available at: https://github.com/UoNMakerSpace/plate-imager-hardware.

#### 2.2.2. Electrical/Control Components

A limitation of the AccelStepper library and relatively low clock speed processors such as the ATmega328P used by the Arduino Uno in the Thermal Imager is motor speed. The maximum steps per second at a clock frequency of 16 MHz is estimated at 4000 but in practice, this is difficult to achieve [17]. To achieve higher motor speeds (and still utilize acceleration and deceleration), the Plate Imager uses a development board with a processor than runs at a much higher clock speed (72 MHz Cortex-M4 microcontroller; Arm Ltd). This board (Teensy 3.2, PJRC) uses 3.3 V signal voltages (rather than the 5 V of the Arduino Uno) but is 5 V tolerant, so some common parts can be used in both systems). Again, Hall-effect sensors are used as limit and home switches. In this design, the sensors are connected at the extremes of the carriage rail and triggered by magnets fitted to the carriage. For this device, 3.3 V-tolerant omnipolar sensors are used. Omnipolar sensors are advantageous in high-speed systems as the sensor will be triggered by the opposite pole of the carriage magnet in the case of an overrun. The stepper motor driver used in the Plate Imager is based on the TB6600 chip (Toshiba) that allows a higher maximum motor current. The microcontroller and stepper driver boards are housed in a 3D-printed enclosure with connectors for power, limit switches and a USB connection to the host computer mounted in the support frame. The motor is powered by a 31 V, 2.4 A power adaptor. A full wiring schematic is given as Appendix A.

#### 2.2.3. Software

To exploit the faster clock frequency of the Cortex-M4 microcontroller, a high-speed driver library (TeensyStep, [25]) was used in the microcontroller sketch software. This allows a theoretical motor speed of 300,000 steps per second with acceleration/deceleration control. User control of experimental parameters (plate diameter, delay between images, save directory) is via a LabVIEW program running on the host PC. This interface also allows monitoring and setting of camera attributes. Images are saved in individual directories for each plate position with unique filenames including acquisition time and date. Experimental settings can be saved as a configuration file and re-loaded on subsequent experimental runs, ensuring that image sets are appended to the same directory. The microcontroller sketch and LabVIEW code are available at https://github.com/UoNMakerSpace/plate-imager-software.

#### 2.2.4. Performance

Characteristics of the Plate Imager are given in Table 3. For comparison, characteristics of a previously published research system [26] and a typical commercially-available actuator are also given.

The Plate Imager outperforms the CPIB Imaging Robot (see Table 2) in all measured parameters. Compared to a research unit for plate imaging based on a leadscrew design [26], the new design has a slightly lower repeatability due to the larger microstep size (Table 3). However, the higher positional accuracy of a leadscrew design results in a slower system and the maximum speed of the Imaging Platform is 20% of that of the Plate Imager, leading to a similar increase in the time required for an experimental run. Leadscrew systems are also relatively expensive—components for the Imaging Platform cost nearly 6 times as much as the Plate Imager (Table 3). Compared to a belt-driven commercial design, the Plate Imager has a smaller minimum microstep size and thus improved repeatability. The commercial model is capable of higher maximum speeds, reflected in an improved temporal resolution. Although the commercial model is cheaper than the leadscrew platform, it is nearly 4 times more expensive than the Plate Imager.

## 3. IoT Environmental Sensor Logger

Network-enabled wireless sensor devices are a rapidly expanding market, found throughout homes [27], businesses, and agriculture [28] and increasingly in research environments [29]. This has led to a number of readily available, low-cost IoT components with a rich ecosystem of hardware components and software libraries. We have taken advantage of this expansion to design a highly modular platform that can host a range of environmental sensors from subdollar to highly-calibrated domain specific sensors costing tens to hundreds of dollars. The platform is also modular with respect to communication platform, designed to operate in its default instance with readily available WiFi but able to be adapted to long-range radio systems (XBee/LoRa/GSM) and deployed into remote locations.

### 3.1. Hardware

The initially developed unit (Figure 6) is a low-cost instantiation of the platform designed to be deployed at high numbers into plant growth facilities in a large academic department. The core of the unit is an ESP32-based microprocessor [30] which allows data from sensors connected via multiple devices busses to be relayed over built-in WiFi hardware. The sensor module is an ultra-low-power unit that measures ambient temperature, relative humidity, barometric pressure and air quality (model BME680 [31]). This sensor can be interfaced with using I2C or SPI serial communication protocols and is widely available on breakout boards to simplify deployment (Figure 6a). The utility of pressure and air quality logging is limited in phenotyping installations and a cheaper sensor (BME280 [32]) is available with similar characteristics for temperature and humidity measurement (Table 4). For comparison, specifications of a commercial standalone sensor and a WiFi-enabled datalogger are also shown.

A printed circuit board (PCB) was designed for deployment of the sensor unit assembly to allow the use of inexpensive, pre-soldered components. The board consists of headers for the ESP32 development board, RTC module, sensor module, and a battery holder for an 18650 LiFePO_4_ battery. Headers are included for a voltage divider circuit to monitor battery voltage, two i2c bus connections for additional sensors and diagnostic unit connection, and serial device headers for connection of future domain specific hardware. Electrical schematics are shown in Appendix A and a populated PCB in Figure 6. Schematics and fabrication files for the PCB are available at https://github.com/UoNMakerSpace/sensor-platform-hardware.

### 3.2. Software

The ESP32 hosts a range of runtime environments with their own system libraries and languages, including JavaScript, Python, Lua, and C++. Our platform is environment and language agnostic, requiring only that the chosen suite can provide a network interface via MQTT(S) and HTTP(S) and can connect to a WiFi network via the security mechanism in place in the monitored environment. The test instantiation is written in C++ using the manufacturer default operating system with Arduino libraries (https://github.com/UoNMakerSpace/psn-node) compiled and uploaded via PlatformIO. The ESP32, like most true IoT units is headless, communicating bidirectionally with host development platforms over serial connections. For this reason, unit administration of the devices, such as maintaining WiFi credentials and server addresses, is generally a laborious task requiring either a development platform or an insecure setup mode accessible with local, private Bluetooth or WiFi networks. We have developed for the platform a simple administration app, written in C# and using WPF libraries (https://github.com/UoNMakerSpace/psn-node-admin), to provide an administration interface for effective, secure, off-line administration. Given the austere locations the units are planned to be deployed into, the variability of wireless communication, and the necessity to conserve battery life, if a connection attempt fails the unit will rapidly timeout, and store the sensor data with the timestamp and send the data on the next successful connection. Units can relay simple debugging messages, such as battery strength, connection signal strength, and number of failed network connections to the server for online monitoring purposes. A simple administration unit can also be connected to the unit for in-field interrogation of debug messages, in the unlikely event of an edge-case connection issue.

### 3.3. Network

Deployed devices relay sensor readings and diagnostic messages in a common, self-describing format using JSON to backend server-side software components (Figure 7). These consist of a message router which processes MQTT PubSub messages and relays them, again using a common described format to a database interface layer, where messages are written into a backend datastore. We have found MQTT to be highly efficient on units with limited processing power, offering reliable probe-driven bi-directional communication, with the backend. In the event of an issue with MQTT communication, the unit can fall back to classic HTTP-POST communication, using the same message format.

The platform does not proscribe the backend datastore, offering flexibility to integrate with existing deployed technologies. For ease of deployment and testability, the initial instantiation combines message routing and data layer into a self-contained unit, backed by a SQL database (https://github.com/UoNMakerSpace/psn-logger, https://github.com/UoNMakerSpace/psn-server). Data are made accessible to the end users by a web server component, written in PHP, which allows a user to interrogate probe data using a web browser. Other dissemination routes are planned, using access to the datastore via REST to provide live feed and notifications via a web or mobile app, or using webservices to provide the logged environmental conditions for phenotyping experiments into integrative stores (IS) such as PHIS [34] and PIPPA [35].

The server-side programs can run on the same hardware, but they can also be modified to allow the message router to run on cheap frontend hardware—for instance, a RaspberryPi as in [36]—while the database and webserver run in either dedicated server hardware, or a low-cost virtual machine either on site or in the cloud. This provides a reliability advantage, with server components running in a datacentre overseen by IT administrators whilst allowing the bespoke IoT-specific functionality to be run closer to the units on the same network (with failover capability), and a cost advantage, as specific server hardware need not be purchased and the frontend is running on low-cost hardware.

### 3.4. Performance

An example sensor log is shown in Figure 8.

In Table 4, the platform, configured with two commonly available temperature sensors (Bosch Sensortech BME680 and BME280, a 1500 mAh LiFePO4 rechargeable battery, and set to 1 min recording interval), is compared to two popular commercially available environmental sensor units: Tinytag Ultra 2 (Gemini Data Loggers TGU-4017) and OM-EL-WIFI-TH-PLUS (Omega Engineering). All the units have similar operating ranges and accuracy, appropriate for their role measuring variables in large environments. The platform developed here shows that despite its low cost, it is competitive with more expensive commercial units in terms of numbers of readings that can be logged and in mode of operation, the trade off with battery life as presented is due to a design requirement for the test unit to log standard experimental runs, which in this case are a few weeks in duration. The run duration is a factor of the logging interval, which, as tested here, is at a high frequency (60 h^−1^)—the unit can perform approximately 30,000 measurements on a standard battery, or 60,000 on a large capacity battery, which would see a lifetime in the several month range at a standard 10 minute logging interval. Minimum reading time is controlled by two figures: the time to wake the unit, read the sensors and determine the median reading for each (approximately 3 s), and a longer period connecting to WiFi and a very rapid delay to relay signals (approximately 0.9 s). The connection to WiFi is a complicated variable—simple secure authentication systems found on home-type routers or WiFi hotspots can be connected to in <5 s (95th percentile), but complex WPA2-Enterprise based systems, such as the academic eduroam system, can take 2-fold longer (90th percentile) or even 4-fold longer (99th percentile) (data not shown). For this reason, to save power, the unit saves readings and connects to WiFi at a regular frequency determined by a user-specified batching number. The unit could be redesigned to step around the delay induced by WiFi connection time with an event based loop, but we do not believe any of these sensors would accurately identify gross environmental parameters in a large monitored space at subminute temporal resolution, and at this sampling frequency any unit would quite rapidly deplete its battery.

Figure 8 shows the performance of the test unit with a BME680 sensor and 5 min logging interval over a week in a glasshouse, along with external measurements at 1 h logging intervals from a calibrated weather station on the same campus. As can be seen, the temperature and humidity sensors perform as expected for an environmental sensor, with little variability, and demonstrate how the glasshouse environment is affected by external conditions.

## 4. Discussion

The vector platforms presented here are inexpensive and easily adapted for multiple use cases. The use of readily-available mechanical and electronic components popularized by the Maker community allows the deployment of bespoke systems at a fraction of the cost of the off-the-shelf platforms. The platforms offer improvements to existing research designs and are comparable in key performance characteristics to commercial models. The modular nature of the designs and the extensive use of 3D-printed components means that the vectors can easily be re-purposed if required.

The sensor platform provides logging of low-cost environmental probes which can be deployed at scale to provide complete fine-granularity coverage over a range of plant phenotyping facilities with designed-in management and administration, and user-targeted distribution of real-time environmental conditions. The platform is low cost, offers comparable features to commercial alternatives, and has been designed to be as modular as possible, while retaining ease of deployment and management, to ensure that it does not restrict deployment to measure any feasible environmental parameter or growth environment.

## Figures and Tables

**Figure 1 sensors-20-03319-f001:**
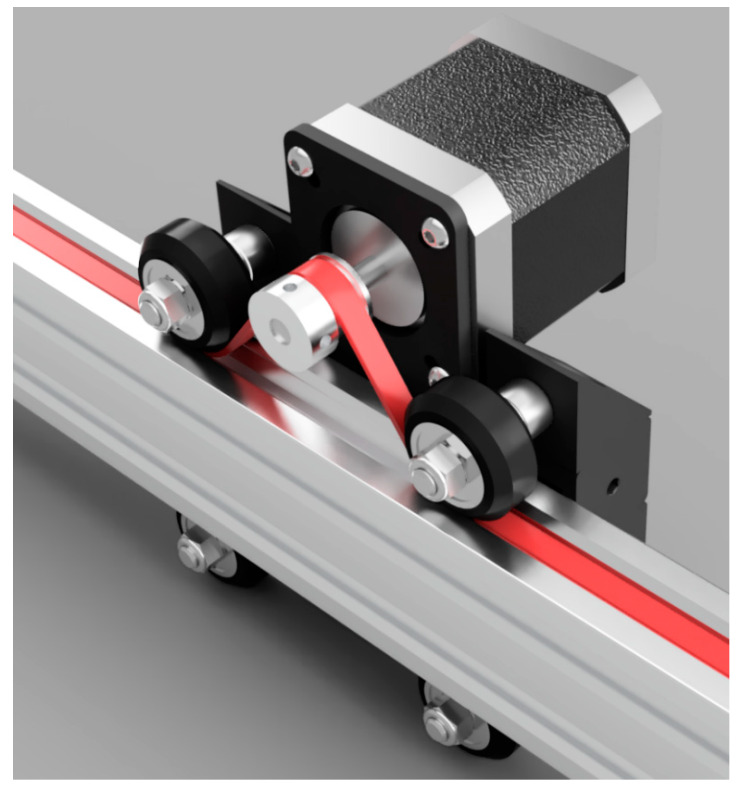
Belt-and-pinion drive system. The timing belt (red) passes under the drive wheels and over the timing pulley to maintain tension in the belt. Drawing files from [10].

**Figure 2 sensors-20-03319-f002:**
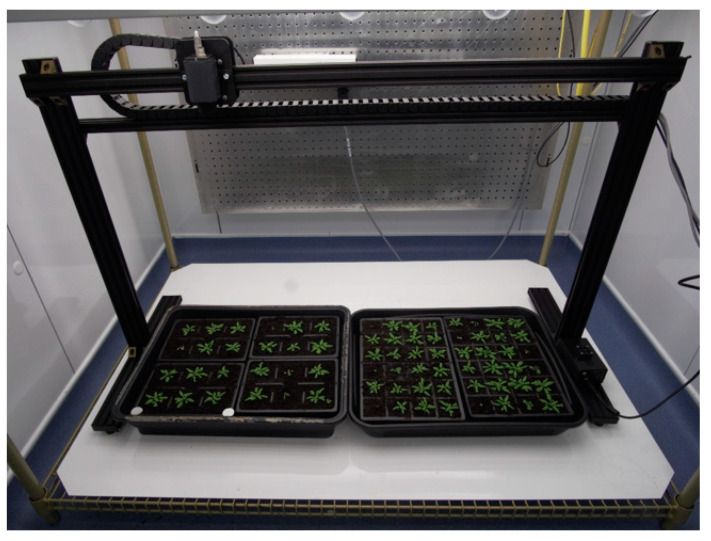
Thermal Imager. For scale, the drive rail is 1.2 m in length.

**Figure 3 sensors-20-03319-f003:**
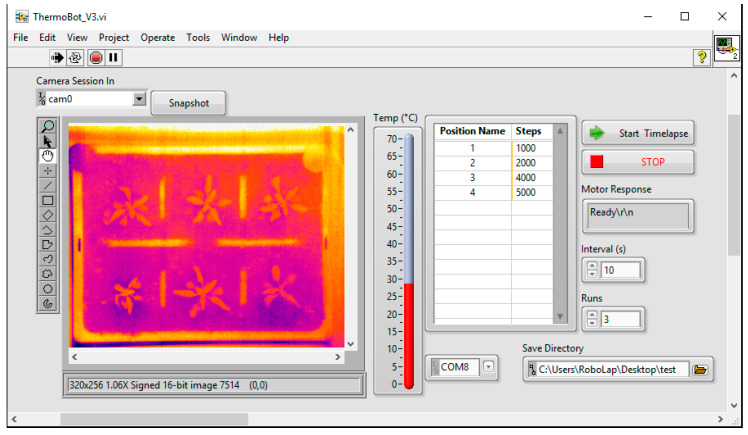
Thermal Imager control software user interface.

**Figure 4 sensors-20-03319-f004:**
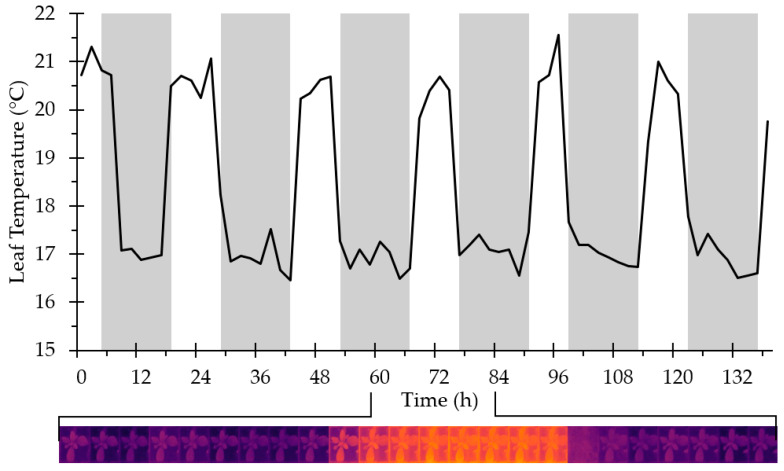
Leaf temperatures of *Arabidopsis thaliana* rosettes recorded using the Thermal Imager. In total, 48 plants were imaged every 30 min for 136 h. For clarity, data for a single plant are shown (mean of three spot measurements) at 2 h intervals. Dark bars show the night photoperiod. Inset: hourly thermographs for the marked 24 h period.

**Figure 5 sensors-20-03319-f005:**
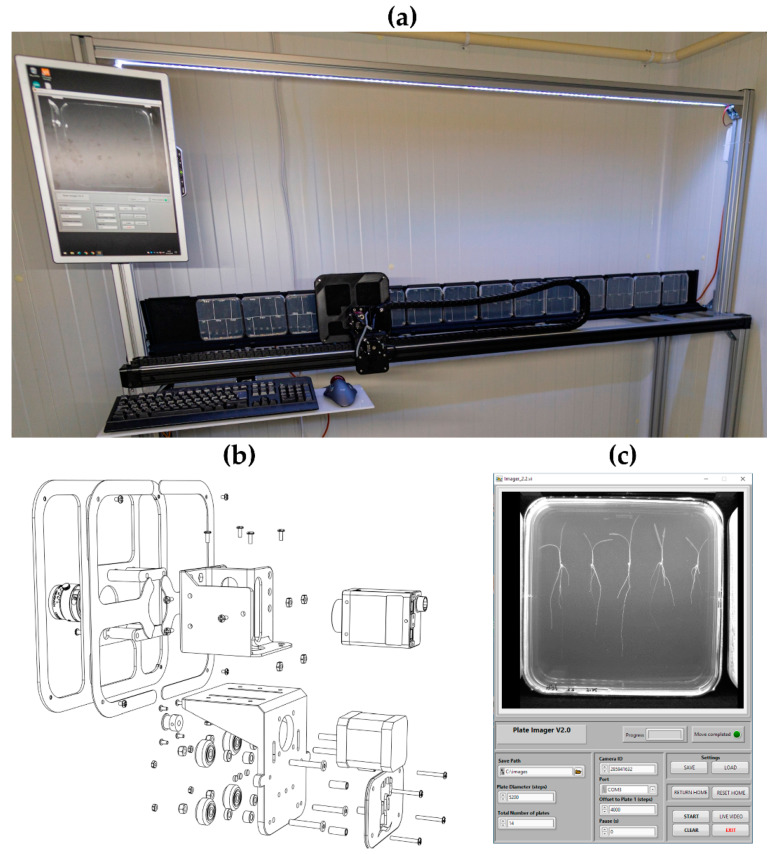
(**a**) Plate Imager robot. (**b**) Carriage plate, sensor holder and baffle assembly. (**c**) Plate Imager control software user interface.

**Figure 6 sensors-20-03319-f006:**
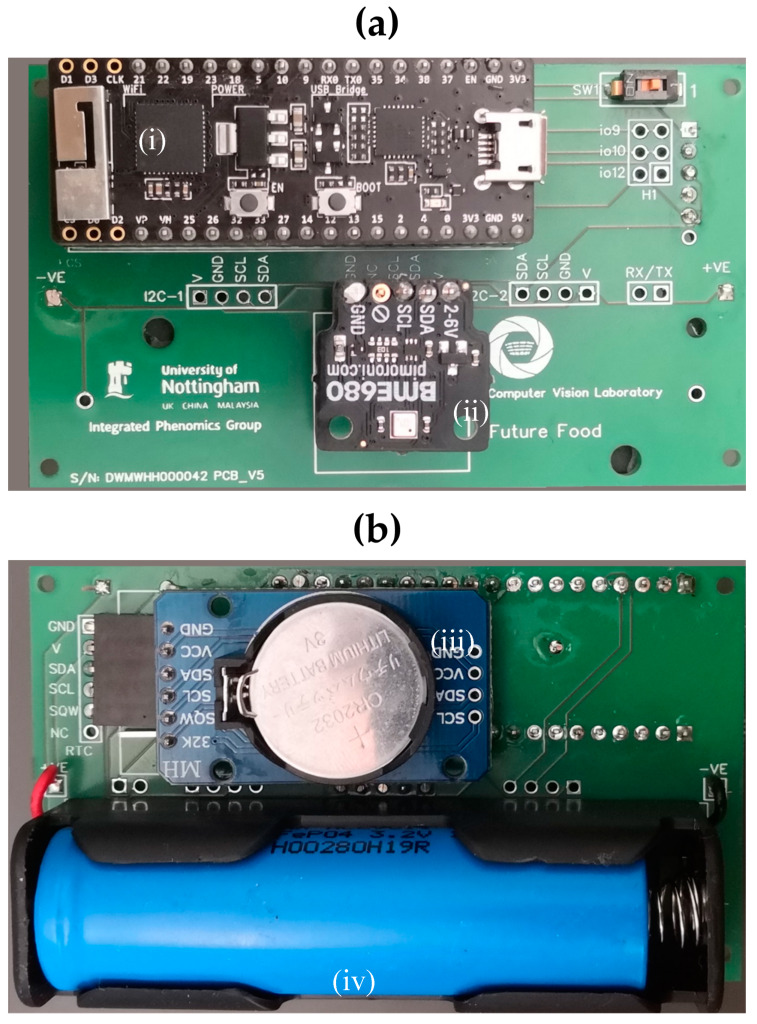
(**a**) Sensor printed circuit board (PCB) top side. (**b**) Bottom side. Main components: (i) ESP32 development board, (ii) sensor daughter board, (iii) real-time clock module, and (iv) LiFePO_4_ rechargeable battery.

**Figure 7 sensors-20-03319-f007:**
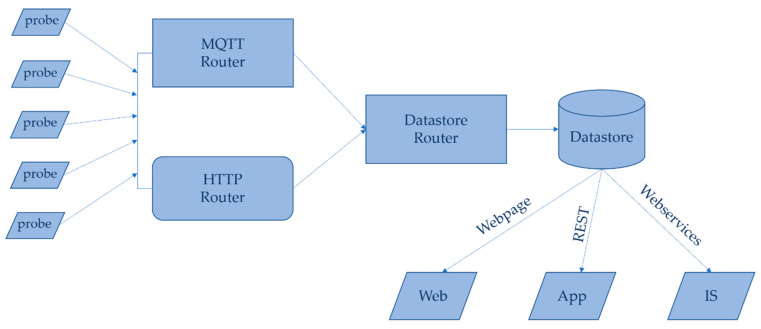
Network architecture of the platform showing flow of measurements and data through the routing layers to end users via web pages, web apps and associated integrative stores (IS).

**Figure 8 sensors-20-03319-f008:**
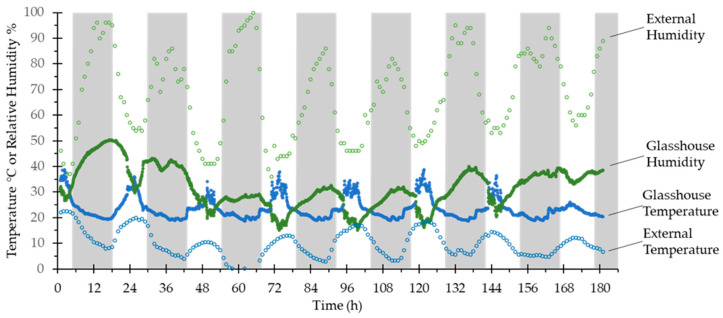
Data recorded by a probe over 182 h in a glasshouse. Green circles are percentage relative humidity, blue circles are temperature (℃). Filled circles are data gathered by the probe in the glasshouse (readings every 5 mins), open circles are from a calibrated weather station approximately 300 m away (readings every 60 mins) for the same period showing external environmental conditions. 0 h was 12 pm on a Saturday afternoon. Dark bars show the period after sunset, light bars the period after sunrise.

**Table 1 sensors-20-03319-t001:** Components for the automated plant phenotyping vectors.

Component	Specifications	Model/Filename	Manufacturer
Thermal Imager			
Stepper motor	Bipolar, 1.8° step angle, 1.68 A/phase	MT-1704HS168A	Motec
Microcontroller	16 MHz ATmega328P	Arduino Uno R3	Arduino
Driver carrier	Arduino shield for removable drivers	CNC Shield V3	Various
Stepper driver	Max 32 microsteps, 2.5 A, 12−40 V	DRV8825	Texas Instruments
Drive belt	7 mm width; 2 mm pitch	GT2-2M	OpenBuilds
Timing pulley	20 teeth; 2 mm pitch	GT2-2M	OpenBuilds
Carriage rail	V-slot profile	40 × 20	OpenBuilds
Carriage wheels	15.2 mm outside diameter (OD), Delrin	Mini V Wheel	OpenBuilds
Carriage plate	3D printed	therm_cm.stl ^1^, therm_cps.stl	UoN ^2^
Sensor holder	3D printed	therm_s_flir.stl	UoN
Electronics box	3D printed	therm_case.stl	UoN
Limit switches	Hall-effect sensor, unipolar, 4.5−24 V	MP101402	Cherry
Sensor	Thermal camera 19 mm lens, 24° field of view	A35 (60 Hz)	FLIR
Plate Imager			
Stepper motor	Bipolar, 1.8° step angle, 1.68 A/phase	MT-1704HS168A	Motec
Microcontroller	72 MHz Cortex-M4	Teensy 3.2	PJRC
Stepper driver	Max 32 microsteps, 3.5 A, 8–45 V	TB6600	Toshiba
Drive belt	7 mm width; 2 mm pitch	GT2-2M	OpenBuilds
Timing pulley	20 teeth; 2 mm pitch	GT2-2M	Openbuilds
Carriage rail	V-slot profile	40 × 20	OpenBuilds
Carriage wheels	15.2 OD, Delrin	Mini V Wheel	OpenBuilds
Carriage plate	3D printed	plate_carriage.stl ^3^	UoN
Sensor holder	3D printed	plate_sm.stl	UoN
Light baffle	3D printed	plate_baffle(1−3).stl	UoN
Electronics box	3D printed	plate_case(1−3).stl	UoN
Limit switches	Hall-effect sensor, omnipolar, 2.5−5 V	AH180	Diodes Inc.
Sensor	FireWire camera, 8 mm lens	Stingray	AVT

^1^ Files available at: https://github.com/UoNMakerSpace/thermal-imager-hardware.

^2^ UoN: 3D printed at the University of Nottingham.

^3^ Files available at: https://github.com/UoNMakerSpace/plate-imager-hardware.

**Table 2 sensors-20-03319-t002:** Comparison of Thermal Imager specifications with competing (research and commercial) robots.

Specification	Thermal Imager	CPIB Imaging Robot [21]	Commercial Actuator ^1^
Drive	Belt and pinion	Toothed belt	Toothed belt
Travel	1.2 m	1.8 m	1.2 m
Step size	200 µm	300 µm	600 µm
Microstep size (minimum)	6.25 µm (32 microsteps)	300 µm (n/a)	9.4 µm (64 microsteps)
Maximum speed	125 mm/s	30 mm/s	5 m/s
Repeatability	~5 µm	0.5 mm	200 µm
Temporal resolution	~40 s/run	~2 min/run	~8 s/run
Cost ^2^	€235	€1060	€3475

^1^ Model ZLW-1660, Igus GmbH.

^2^ Cost excludes camera and host PC.

**Table 3 sensors-20-03319-t003:** Comparison of Plate Imager specifications with competing (research and commercial) robots.

Specification	Plate Imager	CPIB Imaging Platform [26]	Commercial Actuator ^1^
Drive	Belt and pinion	Leadscrew	Toothed belt
Travel	1.5 m	1.5 m	1.495 m
Step size	200 µm	31.75 µm	270 µm
Microstep size (minimum)	6.25 µm (32 microsteps)	0.5µm (64 microsteps)	4.2 µm (64 microsteps)
Maximum speed	300 mm/s	60 mm/s	2000 mm/s
Repeatability	~5 µm	<2 µm	<20 µm
Temporal resolution	68 s/run	~5 mins/run	~20 s/run
Cost ^2^	€780	€4560	€2904

^1^ Model X-BLQ1495-E01, Zaber Technologies, Inc.

^2^ Cost excludes camera and host PC.

**Table 4 sensors-20-03319-t004:** Sensor and logging characteristics.

Specification	BME680	BME280 [32]	TinyTag Ultra ^2^	WiFi Logger ^1^ [33]
Humidity range	0–100% rh	20−80% rh	0 to 95% rh	0–100% rh
Humidity accuracy	±3% rh	±3% rh	±3% rh	±4.0% rh
Humidity response time	8 s	1 s	~10 s	n/a
Temperature range	−40 to 85 °C	−40 to 85 °C	−25 to +85 °C	−20 to 60 °C
Temperature accuracy (25 °C)	±0.5 °C	±0.5 °C	±0.4 °C	±0.8 °C
Minimum time between readings	10^2^ s		1 s	10 s
Maximum readings	250,000 ^3^		32,000	500,000
Battery life	1 month ^4^		1 year	1 year
Online reporting	yes		no	yes
Cost	€19	€11	€135	€133

^1^ Model OM-EL-WIFI-TH-PLUS, OMEGA Engineering, Inc.

^2^ This figure represents the minimum duty cycle time of the unit to read the sensors, store the results and relay over WiFi to a test connection; in network connected mode readings are batched and sent to the network with WiFi authentication time dependent on network configuration (connection to WPA2-Enterprise networks can take up to 20 s).

^3^ This figure is for permanent storage on the unit, acting in logging mode or when not connected to the network.

^4^ This is at a logging interval of 60 s with a 1500 mAh battery.

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
