# Peer review of "Low-Cost Automated Vectors and Modular Environmental Sensors for Plant Phenotyping"

_sensors, 2020, doi:10.3390/s20113319_

Round 1

Reviewer 1 Report

The paper 'Low-cost automated vectors and modular 3 environmental sensors for plant phenotyping" give us an affordable phenotyping system including automated vectors, thermal imager, plate imager and IoT sensors. From the point of practical application, it's a good way for applying these instruments for phenotyping, but the nolvelty is not sufficient from the scientific point.

Their is no precise experiment, no data analysis, and no validation of the whole system.

For phenotyping system, the whole environment parameters, different kinds of sensors, control system and data processing system were needed. I suggest the authors add more sensors, and do an precise experiment for validation of the whole system.

Reviewer 2 Report

Review of "Low-cost automated vectors and modular 2 environmental sensors for plant phenotyping"  by Stuart A Bagley et al.

Submitted to Sensors, May 2020, manuscript n° sensors-814216

This article reports on development and test of 2 imaging systems requiring a more or less rapid or accurate movement of the camera along the scene and a design for autonomous (atmospheric) environmental monitoring and transmission/storage of data. These developments are applied to phenotyping purposes, mainly for growth chamber studies. The designs are well presented and nearly all the documents/information are available to assemble the systems (cf detailed comments).  I was really interested, as a researcher, by these developments and I thank the authors for sharing them with the scientific community. I’m convinced that such (rather) low cost systems will really be useful in in phenotyping studies, but will also inspire developments into other scientific studies necessitating automation of imaging in an affordable manner. If the authors have some feedback on the longer term / intensive use of the developed equipment, concerning reliability, robustness or ease of use with a diversity of users it would be also an added value to the paper (which would not deserve another round of review).

Detailed comments

L80 Table 1  What is the Manufacturer UoN (University of Nottingham ??) => Where are to find the files therm_*.* and plate_*.*  => give web site in the legend.

L97 and L128  https://github.com/UoNMakerSpace/thermal-imager-hardware (and -softaware) => Error404 , no documents  accessible on May 25, 2020

L168 and L201: https://github.com/UoNMakerSpace/plate-imager-hardware  and  https://github.com/UoNMakerSpace/plate-imager-software => Error404 , no documents  accessible on May 25, 2020

Reviewer 3 Report

This a very well writen paper describing the details of glasshouse equipment including thermal camera monitoring the temperature of the plant leaves. Moreover other sensors are also monitored such as ambient temperature, hummidity etc. The information included in this paper is very useful for the construction of robotic vectors doing the measurements referenced above. The quality of the the developed prototypes is high. All the implementation details are available to the reader, from construction details and mechanical part dimensions to the developed software that is referenced in its repository  

My main concern is that the nature of this paper matches a manual rather than a scientific paper. All the mechanical and development details that are supplied by the authors are very useful to an application user but a theoretical background is missing. 

To be more specific there are no results compared to some ground truth in order to assess the advantages of the proposed system compared to other similar approaches. I will try to select what features could be the ones to be compared e.g., a) the temperature, humidity precision of the developed sensors, b) the position of the moving camera, c) productivity improvements, d) cost savings, e) speed in detecting abnormal situations in the plants, etc.  

In this sense the references could be extend to include other similar approaches that could be compared to the proposed one. 

Of course I also take into consideration that the paper is submitted to Sensors journal that primarily accepts papers that describe in detail new sensor structures and thus merely describing the developed sensors may be adequate, this will be judged by the editors. However, I believe that including comparison of features like the ones described above would be useful.

Round 2

Reviewer 1 Report

The authors made some compariosn with existed sensors, and the manuscript was improved a lot from the scientific aspect. I have no more comments.

Reviewer 3 Report

The authors have added some comparison at 3 tables and extended the text accordingly. Taking into consideration the high quality of the presentation that I also recognized from the 1st version of the paper I suggest that this paper should be published